

# Changes in red blood cell parameters during incremental exercise in highly trained athletes of different sport specializations

Monika Ciekot-Sołtysiak[1], Krzysztof Kusy[1], Tomasz Podgórski[2], Barbara Pospieszna[1] and Jacek Zieliński[1]

[1] Department of Athletics Strength and Conditioning, Poznan University of Physical Education, Poznan, Poland
[2] Department of Physiology and Biochemistry, Poznan University of Physical Education, Poznan, Poland

Corresponding author
Monika Ciekot-Sołtysiak,
ciekot@awf.poznan.pl

## ABSTRACT

**Background**. During physical exercise, the level of hematological parameters change depending on the intensity and duration of exercise and the individual's physical fitness. Research results, based on samples taken before and after exercise, suggest that hematological parameters increase during incremental exercise. However, there is no data confirming this beyond any doubt. This study examined how red blood cell (RBC) parameters change during the same standard physical exertion in athletes representing different physiological training profiles determined by sport discipline.

**Methods**. The study included 39 highly trained male members of national teams: 13 futsal players, 12 sprinters, and 14 triathletes. We used multiple blood sampling to determine RBC, hemoglobin (Hb), hematocrit value (Hct), mean corpuscular volume (MCV), mean corpuscular hemoglobin (MCH), mean corpuscular hemoglobin concentration (MCHC), and red blood cell distribution width (RDW) before, during (every 3 min), and after (5, 10, 15, 20, and 30 min) an incremental treadmill exercise test until exhaustion.

**Results**. There were no significant exercise-induced differences in RBC parameters between athletic groups. No significant changes were recorded in RBC parameters during the low-intensity phase of exercise. RBC, Hb, and Hct increased significantly during incremental physical exercise, and rapidly returned to resting values upon test termination.

**Conclusions**. The general pattern of exercise-induced changes in RBC parameters is universal regardless of the athlete's physiological profile. The changes in RBC parameters are proportional to the intensity of exercise during the progressive test. The increase in hemoglobin concentration associated with the intensity of exercise is most likely an adaptation to the greater demand of tissues, mainly skeletal muscles, for oxygen.

## INTRODUCTION

Red blood cells (RBC; erythrocytes) carry out several important functions, especially during exercise. First of all, RBC play a crucial role in oxygen ($O_2$) transport from the lungs, through the arteries to the contracting muscles. Moreover, the release of adenosine triphosphate (ATP) and nitric oxide (NO) from RBC contributes to the dilation of blood vessels and enhances blood flow through working muscles. Hemoglobin (Hb) contributes to buffering pH changes in blood by combined transport of carbon dioxide ($CO_2$) and hydrogen ions ($H^+$) in form of bicarbonate ions ($HCO_3^-$). All of this requires an adequate RBC count (*Mairbäurl, 2013*).

### Hematological parameters after exercise

Very short but intense exercise, lasting from a few seconds to three minutes, increases the hematocrit value (Hct) and Hb concentration by about 6–8% (*Freund et al., 1987*). Exercise performed on a treadmill or bicycle ergometer, lasting longer than ten minutes, causes an increase in Hb concentration, RBC count, and Hct by 4–10% (*Freund et al., 1987*; *Laub et al., 1993*; *Neuhaus & Gaehtgens, 1994*). In the bloodstream of a healthy person, the values of these three hematological parameters are significantly higher after intense physical exercise than at rest (*Davidson et al., 1987*; *Szygula, 1990*). It is mainly the result of dehydration, high blood viscosity, and fluid displacement within intracellular and extracellular spaces. During the post-exercise recovery period, erythrocyte system indices return to baseline levels, and sometimes continue to drop below resting values (*Davidson et al., 1987*; *Freund et al., 1987*; *Laub et al., 1993*; *Neuhaus & Gaehtgens, 1994*; *Szygula, 1990*).

The magnitude of the exercise-induced changes in hematological parameters depends on current external conditions (temperature, humidity), the intensity and duration of exercise, and the level of physical fitness of the examined person. Lower Hct was observed in athletes than in healthy untrained individuals (*Sharpe et al., 2002*). High Hct rises blood viscosity, which may put more stress on the heart and, consequently, increase the risk of cardiac overload (*Mairbäurl, 2013*). Athletes usually have a higher plasma volume (PV) than sedentary people (*Heinicke et al., 2001*), and this is the main cause of their lower Hct values. Oxygen transport to working muscles in highly trained individuals depends, among other things, on rapid diastolic filling of the heart, which is possible with a sufficiently large blood volume (*Levine, 2008*). A high total mass of RBC and Hb in trained individuals, with a sufficiently large blood volume (BV), increases the $O_2$ transport capacity (*Schmidt et al., 2010*).

### Hematological parameters in athletes from various sports specializations

Athletes competing at the highest level differ in terms of their dominant metabolic profile dictated by their sport. Speed-power sports (*e.g.*, sprint, strength sports) with very high-intensity efforts, require mainly energy derived from anaerobic metabolism (*Duffield, Dawson & Goodman, 2004*). On the other hand, endurance sports (*e.g.*, long–distance running, triathlon) are characterized by a predominance of aerobic metabolism (*Billat et al., 2001*). In most team sports and many individual sports, where the workloads depend

on interactions with partners and/or opponents, the energy for muscle activity is obtained from both aerobic and anaerobic pathways (mixed-trained sports).

Some research results suggest that the type of training load may be correlated with the level of blood indices at rest. Athletes from different sport specializations are characterized by different hematological parameters. For example, triathletes had significantly higher Hb (*Pospieszna et al., 2019*; *Zieliński & Kusy, 2012*) and Hct (*Pospieszna et al., 2019*) resting levels throughout the training season than sprinters, but no such differences were not shown between endurance- and mixed-trained athletes (*Pospieszna et al., 2019*). Interestingly, males practicing strength sports were observed to have higher Hb, Hct, and RBC resting values compared to endurance athletes (*Schumacher et al., 2002*). Resting blood volume in athletes with dominant anaerobic activity was significantly lower than in the endurance groups (*Heinicke et al., 2001*). Several studies did not demonstrate any differences in hematological parameters between endurance, speed-power, and mixed elite athletes (*Ciekot-Sołtysiak et al., 2018*; *Malczewska-Lenczowska et al., 2013*; *Zarebska et al., 2018*).

## Hematological parameters and intensity of exercise

The magnitude of changes in hematological parameters may depend on the intensity of exercise. Moderate-intensity swimming training in rats increased Hb and Hct, while intense training gave the opposite effect (*Liu et al., 2006*). Earlier studies indicated a decrease in PV after heavy endurance exercise in humans (*Ahmadizad & El-Sayed, 2003*; *Ahmadizad & El-Sayed, 2005*; *Freund et al., 1987*; *Laub et al., 1993*; *Neuhaus & Gaehtgens, 1994*). On the other hand, strength training did not affect changes in plasma volume in sedentary young and middle-aged males (*McCarthy et al., 1997*). These outcomes point out that the change in PV after endurance training may be greater than that resulting from other training modalities. RBC, Hb, and Hct increased after moderate-intensity training, but not after high-intensity intermittent training in sedentary young adults (*Findikoglu et al., 2014*). So far, no studies are available on exercise-induced changes in hematological parameters in elite athletes engaged in anaerobic sports. Differences in training loads specific to a particular sports discipline may cause ambiguous results regarding changes in hematological parameters in different groups of athletes. Different patterns of changes in resting hematological parameters during the one-year training cycle in athletes (*Ciekot-Sołtysiak et al., 2018*) suggest that particular training periods, related to the physiological adaptation of athletes, should be taken into account when interpreting blood results.

While reviewing the literature, we did not find any studies describing how RBC parameters change during the same standard physical exertion in athletes representing different physiological training profiles determined by sport discipline. Also, the changes in hematological parameters in athletes were only based on blood samples taken before and after exercise, ignoring intensity-related changes during exercise. Research results based on samples taken before and after exercise suggest that hematological parameters increase during incremental exercise. However, there is no data confirming this beyond any doubt and the pattern of the changes is not known. Most of the studies included endurance athletes, such as runners (*Carin et al., 2023*; *Davidson et al., 1987*; *Nader et al., 2018*; *Neuhaus, Behn & Gaehtgens, 1992*; *Reinhart, Staeubli & Straub, 1983*; *Robert et al., 2020*;
*Tripette et al., 2011*) or cyclists (*Nader et al., 2018*; *Simmonds, Connes & Sabapathy, 2013*), and were usually performed during competitions without standardizing the conditions. Other studies examined the effects of different types of training on hematological parameters in recreational athletes (*Ahmadizad & El-Sayed, 2005*; *Bizjak et al., 2020*). In this study, we aimed to examine highly trained athletes in standardized laboratory conditions to limit the effects of confounding factors. Our highly trained athletic groups were stimulated for a long time with training loads dominated by aerobic, anaerobic, or mixed energy metabolism. We measured all parameters during the same phase of the annual training cycle for all athletes and used multiple blood sampling during exercise.

This study aims to examine how RBC parameters change during exercise of increasing intensity in competitive athletes of different specialties. We hypothesize that RBC parameters will increase proportionally to the intensity of incremental exercise in highly trained male athletes.

## MATERIALS & METHODS

### Subjects

The study included 39 highly trained male athletes from sport disciplines representing different training profiles and physiological adaptations: speed-power sports (sprinters; $n = 12$; aged $24.5 \pm 3.2$ years), endurance sports (triathletes, $n = 14$; aged $22.7 \pm 4$ years), and mixed-trained sports (futsal players, $n = 13$; aged $25.7 \pm 4.3$ years). All athletes were members of the Polish national teams with competitive sport history longer than 5 years (FU = $10.4 \pm 4$ years; SP = $9.7 \pm 2.5$ years; TR = $9.1 \pm 2.2$ years). Futsal players who participated in our research played offensive, defensive, or universal positions (goalkeepers were excluded). Sprinters were specialized in the 100 m, 200 m, and 4 ×100 m relay events. Triathletes were specialized in the standard Olympic distance: 1.5 km swimming, 40 km cycling, 10 km running. All measurements were performed at the beginning of the preparatory phase of the athletes' annual training cycle. None of the athletes were tested positive for any illegal performance enhancing drugs.

### Study design

Subjects were instructed not to participate in any training sessions at least 24 h before testing. At the beginning, the participants were fully informed of the purpose and risks of the study, signed the written consent and filled out all required forms. All tests were conducted at the Human Movement Laboratory of the Poznan University of Physical Education (in laboratory conditions with standardized temperature and humidity) in the morning. Subjects underwent body composition analysis and venous blood collection at rest and then an incremental exercise test on a mechanical treadmill (H/P Cosmos Pulsar; Sports & Medical, Nussdorf-Traunstein, Germany). Initial speed was set at 4 km · h$^{-1}$ and increased after 3 min to 8 km · h$^{-1}$. After that point, treadmill speed increased by 2 km · h$^{-1}$ every 3 min. An incremental exercise test was performed until participants reached volitional exhaustion. They were monitored for next 30 min of post-exercise restitution. Fluid intake was not allowed before, during and 30 min after the test. The project was approved by the Ethics Committee at the Karol Marcinkowski Poznan University of

Medical Sciences (Decision No 1252/18) and has been performed according to the ethical standards laid down in the Declaration of Helsinki. This study was registered in the ClinicalTrials.gov, identifier number: NCT05672758.

## Somatic variables

Height and total body mass were measured using digital stadiometer (SECA 285; SECA, Hamburg, Germany). The subjects' body composition (total skeletal muscle, fat and lean mass) were measured using the dual X-ray absorptiometry method (DXA, Lunar Prodigy; GE Lunar Healthcare, Madison, WI, USA).

## Blood collection

Blood samples were obtained from the antecubital vein *via* peripheral venous catheter (1.3× 32 mm, BD Venflon Pro, Becton Dickinson, Helsingborg, Sweden) into syringes comprising EDTA (S-Monovette, 2.7 ml KE; Sarstedt, Nümbrecht, Germany) by the medical staff, at rest, at the end of each 3-min stage above $10 \text{ km} \cdot \text{h}^{-1}$, immediately after exercise and in the 5th, 10th, 15th, 20th, and 30th min of post-exercise recovery.

## Physiological and biochemical parameters

Respiratory parameters were measured (breath by breath) during an incremental exercise test by the MetaLyzer 3B ergospirometer and analyzed using the MetaSoft Studio 5.1.0 software package (Cortex Biophysik GmbH, Leipzig, Germany). First and second ventilatory threshold (VT1 and VT2) were estimated from breath-by-breath data. Lactate concentration (Biosen C-line; EKF Diagnostics, Barleben, Germany), and creatine kinase activity (Reflotron Plus, Roche, Switzerland) were immediately measured in capillary blood.

## Hematological parameters

The analysis of blood samples was carried out on an automated hematology analyzer Sysmex XS-1000i device (Sysmex Europe, Hamburg, Germany). Erythrocyte count (RBC), hemoglobin concentration (Hb), hematocrit value (Hct), red blood cell indices (MCV, MCH, MCHC), and red blood cell distribution width (RDW) were determined.

## Plasma volume variation

Changes in plasma volume (% ΔPV) during exercise were calculated from the equation:

$$\%\Delta PV = [10000 \cdot (Hct_x - Hct_y)] \cdot [Hct_y \cdot (100 - Hct_x)]^{-1}$$

where $Hct_x$ is resting value, and $Hct_y$ are values during exercise (*Galy et al., 2014*).

## Statistical analyses

The sample size was estimated based on the assumption that effect size will be at least medium. Using an $\alpha$-level of 0.05, a statistical power of 0.80, and a nonsphericity correction of 0.70, it was calculated that at least seven individuals in a single group would be needed to detect significant differences in changes in RBC parameters during exercise in athletes representing different physiological training profiles (G*Power 3.1.9.7; Heinrich-Heine-Universität Düsseldorf, Germany). To test the sports discipline effect, the exercise/recovery

effect, and their interaction, the two-way analysis of variance (ANOVA) was performed. A one-way ANOVA was used to determine differences between the basic and resting characteristics of the studied groups. If significant main effects were found ($p < 0.05$), Scheffe post hoc adjustments were used. All effect sizes for ANOVA were expressed as $\eta^2$ and defined as small (0.01), medium (0.06), or large (0.14). The statistical analyses were carried out using STATISTICA 13.3 software (Tibco Software, Inc., Palo Alto, CA, USA).

## RESULTS

### Subjects description

No significant differences were observed between futsal players, sprinters, and triathletes in terms of age, training history, height, resting creatine kinase activity, and resting and post-exercise lactate concentration. Sprinters presented the highest skeletal muscle mass, total lean mass, but the lowest total fat mass. Triathletes were characterized by the lowest total body mass, BMI, and the highest maximal oxygen uptake (VO$_2$max). Futsal players had a significantly greater total fat mass than other groups. The most significant differences between the athletes were found in body composition components (Table 1). Red blood cells count, Hb concentration and Hct values increased progressively during exercise, reaching the highest value during maximum exertion, and then a linear decrease was observed (see Fig. 1). No significant changes in other red blood cell indices (mean corpuscular volume, MCV; mean corpuscular hemoglobin, MCH; mean corpuscular hemoglobin concentration; MCHC) and red blood cell distribution width (RDW) were noted during exercise and recovery period (see Fig. 2).

### Red blood cells count

Red blood cell count (see Fig. 1A) at rest, during the incremental exercise test, at the point of exhaustion and during recovery did not differ between the groups ($p = 0.99$, $\eta^2 = 0.01$). After applying post-hoc correction, no statistically significant difference was obtained for the Stage*Group interaction effect. During incremental exercise test, RBC increased progressively, being significantly higher in the 18th minute of the run (14 km $\cdot$ h$^{-1}$) in futsal players (4.8%, $p < 0.01$) and in the 21st minute of the run (16 km $\cdot$ h$^{-1}$) in sprinters and triathletes (5% and 5.2% increase, respectively, $p < 0.001$ both). All athletes reached their peak RBC count at maximum intensity at the end of the test, and maximum values were significantly different from resting values in all athletic groups (increase by 8.2% in futsal players, by 7.7%, in triathletes, and by 5.2% in sprinters, in relation to resting values, $p < 0.001$ in each group). The first significant decrease in RBC count from the maximum values was observed 5 min after the end of the test in futsal players (4.3%, $p < 0.01$), after 10 min in triathletes (4.3%, $p < 0.05$) and after 15 min in sprinters (5.5%, $p < 0.01$). There were no significant differences between resting RBC and any recovery RBC counts (5th, 10th, 15th, 20th, and 30th min of post-exercise recovery) in any of the groups.

### Hemoglobin concentration

No significant differences in Hb concentrations between the groups were found ($p = 0.08$, $\eta^2 = 0.13$). After calculating post-hoc tests, no statistically significant difference was

**Table 1  Basic characteristics of the athletic groups.**

| | Futsal players ($n = 13$) | Sprinters ($n = 12$) | Triathletes ($n = 14$) | ANOVA[*] ($\eta^2$ effect size) |
|---|---|---|---|---|
| Age (yr) | 25.7 ± 4.3 | 24.5 ± 3.2 | 22.7 ± 4.0 | 0.15 (0.10) |
| Training history (yr) | 10.4 ± 4.0 | 9.7 ± 2.5 | 9.1 ± 2.2 | 0.52 (0.10) |
| Height (cm) | 181.9 ± 6.2 | 185.7 ± 4.8 | 182 ± 5.5 | 0.17 (0.03) |
| Body mass (kg) | 79.6 ± 9.3a | 82.7 ± 6.1a | 73 ± 6.9#§ | 0.007 (0.24) |
| BMI (kg m$^{-2}$) | 24.1 ± 2.3a | 24.0 ± 1.1a | 22.0 ± 1.7#§ | 0.008 (0.24) |
| SMM (kg) | 35.1 ± 4.0§ | 41.0 ± 4.6#a | 32.8 ± 3.2§ | <0.001 (0.44) |
| Total bone mass (kg) | 3.6 ± 0.3 | 3.7 ± 0.4a | 3.2 ± 0.4§ | 0.004 (0.26) |
| Total fat mass (kg) | 12.9 ± 3.1§a | 8.6 ± 1.2# | 10.0 ± 1.8# | <0.001 (0.4) |
| Total lean mass (kg) | 63.7 ± 6.7§ | 71.5 ± 6.3#a | 60.4 ± 5.4§ | <0.001 (0.38) |
| CK (U l$^{-1}$) | 275.0 ± 110.2 | 357.6 ± 159.5 | 223.2 ± 127.2 | 0.05 (0.16) |
| La$_{rest}$ (mmol l$^{-1}$) | 0.9 ± 0.2 | 0.9 ± 0.2 | 1.0 ± 0.3 | 0.84 (0.01) |
| La$_{max}$ (mmol l$^{-1}$) | 10.5 ± 2.5 | 10.0 ± 1.3 | 10.3 ± 2.2 | 0.80 (0.01) |
| La$_{R30}$ (mmol l$^{-1}$) | 3.8 ± 1.5 | 2.9 ± 1.1 | 3.7 ± 1.7 | 0.29 (0.06) |
| VO$_2$max (ml kg$^{-1}$ min$^{-1}$) | 57.1 ± 3.1a | 53.4 ± 4.1a | 69.5 ± 4.1#§ | <0.001 (0.78) |

**Notes.**

[*]One-way ANOVA. Values are means ± SD.

[#]Significantly different from futsal players.

[§]Significantly different from sprinters.

[a]Significantly different from triathletes.

Abbreviations: BMI, body mass index; SMM, skeletal muscle mass; CK, creatine kinase activity; La$_{rest}$, resting lactate concentration; La$_{max}$, maximal lactate concentration; La$_{R30}$, lactate concentration in 30th min of recovery period; VO$_2$max, maximal oxygen uptake.

observed for the Stage*Group interaction effect. The pattern of changes in Hb (Fig. 1B) during exercise were similar to the pattern of changes in RBC count. The first significant increase in Hb concentration from the resting values was observed in the 15th minute of the run (12 km · h$^{-1}$) for futsal players (4.4%, $p < 0.05$) and in the 21st minute of the run (16 km · h$^{-1}$) for sprinters and triathletes (5.2% and 5.3% increase, respectively, $p < 0.001$ both). The highest Hb values were achieved at the maximum intensity at the end of the exercise (increase by 8.5% in futsal players, in triathletes by 8.1% in triathletes, and by 6% in sprinters from resting values, $p < 0.001$ all). After maximum exertion, Hb decreased progressively and was significantly lower in the 5th min in futsal players (4%, $p < 0.05$), in 10th min in triathletes (4%, $p < 0.05$), and in 15th min of post-exercise recovery in sprinters (5.6%, $p < 0.001$). Resting Hb did not differ from values obtained in the post-exercise recovery phase in any of the groups.

## Hematocrit

For Hct values (Fig. 1C), no significant between-group differences were observed during exercise and recovery ($p = 0.25$, $\eta^2 = 0.07$). After applying post hoc correction, no statistically significant difference was obtained for the Stage*Group interaction effect. Hematocrit increased progressively during exercise. The first difference in Hct appeared in futsal players in the 18th minute of the run (14 km · h$^{-1}$; 4.8% increase, $p < 0.05$). In sprinters and triathletes, the first significant increase was after the 21st minute of exercise (16 km · h$^{-1}$; 5.4% both, $p < 0.001$ both). Hematocrit reached its highest value during

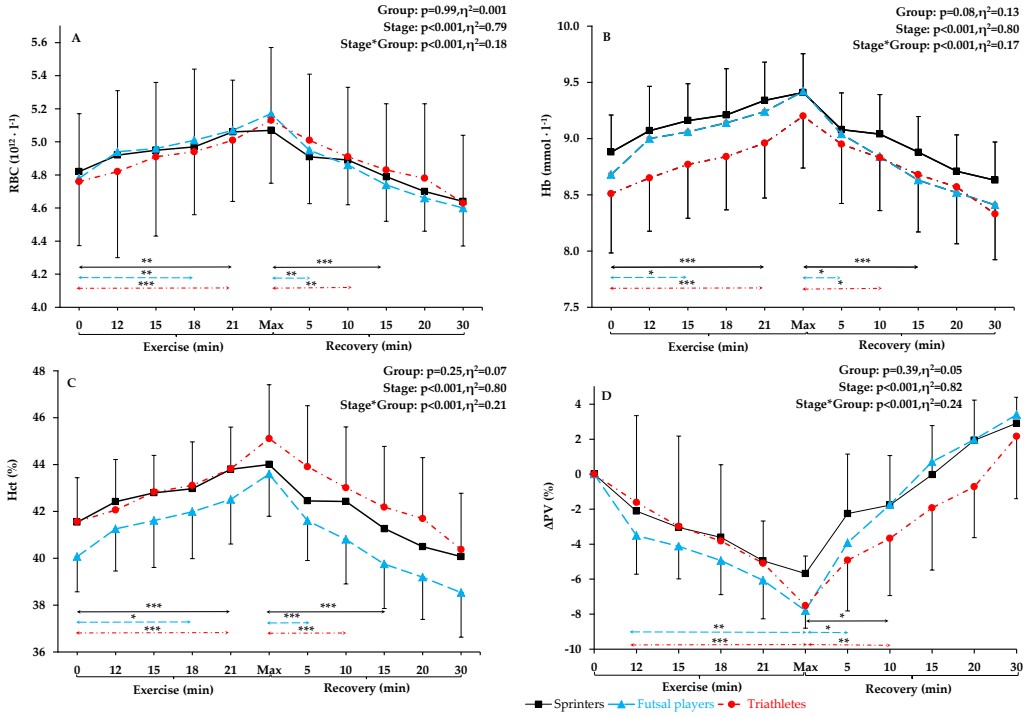

**Figure 1** RBC, Hb, Hct, and Δ. PV level before exercise, during incremental treadmill test until exhaustion, and post-exercise recovery in sprinters, futsal players, and triathletes. (A) Red blood cells (RBC), (B) hemoglobin (Hb), (C) hematocrit (Hct), and (D) plasma volume variation (Δ PV). Values are means ± SD. Horizontal arrows indicate the first significant differences from samples taken at rest and maximal exercise within groups. Significant differences between blood sampling points: ***$p < 0.001$, **$p < 0.01$, *$p < 0.05$.

maximum exertion (increase by 8.8% in futsal players, by 8.5% in triathletes, and by 5.9%, in sprinters compared to resting values, $p < 0.001$ in each case). Thereafter, a return to resting values was observed. After maximal exertion, Hct gradually decreased and was significantly lower in the 5th min in futsal players (6.4%, $p < 0.05$), in the 10th min in triathletes (4.7%, $p < 0.001$) and in the 15th minute in sprinters (6.2%, $p < 0.001$). Each value obtained after exercise did not differ from resting Hct in any of the groups.

## Plasma volume variation

Significant between-group differences in plasma volume variation (Fig. 1D) were not observed. After applying post-hoc tests, no statistically significant difference was observed for the Stage*Group interaction effect. There was a statistically significant decrease in plasma volume variation at the end of the exercise in triathletes ($p < 0.001$) and futsal players ($p < 0.01$).

## First and second ventilatory threshold

The first ventilatory threshold (VT1) occurred between 10 and 12 km · h$^{-1}$ for futsal players and sprinters, and between 12 and 14 km · h$^{-1}$ for triathletes. The second ventilatory
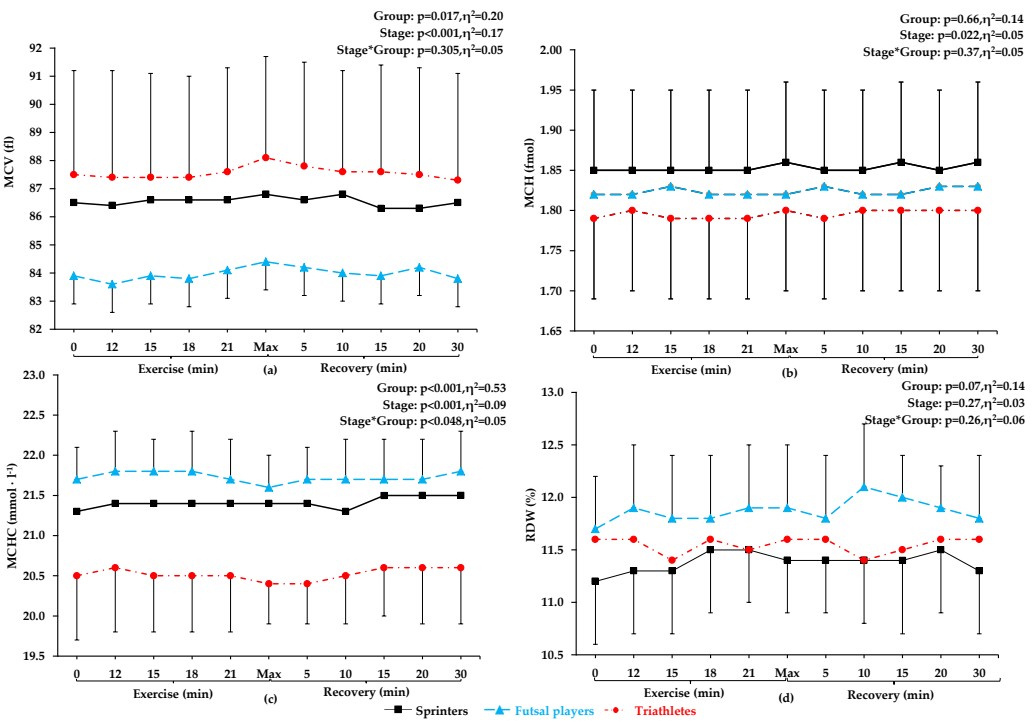

**Figure 2 MCV, MCH, MCHC, and RDW before exercise, during incremental treadmill test until exhaustion, and post-exercise recovery in sprinters, futsal players, and triathletes.** Mean corpuscular volume (MCV), mean corpuscular hemoglobin (MCH), mean corpuscular hemoglobin concentration (MCHC), and red blood cell distribution width (RDW). Values are means±SD. After applying post-hoc correction, no statistically significant difference was obtained for the group and stage interaction effect.

threshold (VT2) occurred between 14 and 16 km · h$^{-1}$ for futsal players and sprinters, and above 16 km · h$^{-1}$ for triathletes.

## DISCUSSION

This study examined the change in RBC parameters and Hct during incremental exercise in highly trained athletes representing different training profiles and physiological adaptations: sprinters, triathletes, and futsal players. Each group was characterized by a different physiological training profile. During competition, sprinters use mostly anaerobic energy sources for muscle activity and related processes, triathletes mainly relay on aerobic metabolism, and futsal players use both aerobic and anaerobic pathways in a more balanced way.

Our main findings are that (i) RBC count, Hb concentration, and Hct values increased gradually during exercise, reaching the highest value during maximum exertion, and then decreased to pre-exercise levels, (ii) there are no significant exercise induced differences in RBC parameters between athletes representing distinct physiological training profiles, (iii) significant changes in RBC parameters were recorded during the high-intensity phase of the incremental exercise test, but not at lower intensities, (iv) RBC, Hb and Hct returned to resting values during the 30-min recovery period, and (v) the general pattern

Ciekot-Sołtysiak et al. (2024), *PeerJ*, DOI 10.7717/peerj.17040                                         9/18

of exercise-induced changes in RBC parameters is universal regardless of the athlete's physiological profile.

## Hematological parameters during exercise

This study confirmed our hypothesis that RBC parameters will increase proportionally to the intensity of incremental exercise in highly trained male athletes of different sport specialization. Red blood cells count, Hb concentration, and Hct values increased progressively during exercise, reaching the highest value during maximum exertion, and then a decrease was observed. Simultaneously, no significant changes in red blood cell indices (MCV, MCH, MCHC) and red blood cell distribution width (RDW) were noted during exercise and recovery period.

At the beginning of exercise, a slight increase (1–2%) in Hct and Hb may be caused by the ejection of blood from the capillaries of previously inactive muscles (*Watts, 1992*). Moreover, *Laub et al. (1993)*, noted that erythrocytes released from contracting spleen during intense exercise may also be responsible for an increase in hematocrit, but taking into the consideration the relatively small size of the human spleen this rise is lower than 2% (*Shephard, 2016*). During exercise, there is a reduction in plasma volume by 10–20%, which causes hemoconcentration (*El-Sayed, Ali & El-Sayed Ali, 2005*). Such a significant reduction in plasma volume during short-term exercise cannot be the result of excessive sweating. Instead, there is a phenomenon of water moving from the vessels to the extravascular space. Increases in hydrostatic pressure in capillaries, in the filtration surface (result of blood redistribution to vascular areas previously poorly perfused), in the stroke volume, and in osmotic pressure in the muscle tissue (result of osmotically active metabolites of working muscles) are the main reasons for the displacement of fluid from intra- to extravascular space (*Connes et al., 2013*). In the case of plasma volume, hydration control is important. Unfortunately, we have not evaluated the athletes' hydration level and sweating rate during exercise. During the study, the athletes stayed in laboratory conditions with standardized temperature and humidity. Fluid intake was not allowed before, during and 30 min after the test.

The observed changes in Hct values are consistent with one previous study, where a similar rise in Hct was demonstrated after exercise of similar intensity in endurance-trained athletes (*Nader et al., 2018*) and after acute exercise in recreationally trained and untrained males (*Ahmadizad & El-Sayed, 2005*; *Freund et al., 1987*; *Laub et al., 1993*; *Neuhaus & Gaehtgens, 1994*). Our results are also consistent with the results of previous studies regarding hemoglobin concentration (*Ahmadizad & El-Sayed, 2005*; *Davidson et al., 1987*; *Freund et al., 1987*; *Nader et al., 2018*; *Neuhaus & Gaehtgens, 1994*; *Szygula, 1990*) and red blood cells count (*Ahmadizad & El-Sayed, 2005*) after exercise. So far, there has been a lack of studies on exercise-induced RBC parameter changes in futsal players and sprinters.

The increase in RBC count, Hb concentration, and Hct values during incremental exercise in our athletes may be partly explained by fluid shifts between vessels and extravascular space. We noted a decrease in plasma volume in all athletes at the point of exhaustion. These results are in line with earlier studies indicating a decrease in plasma volume of about 10% after heavy exercise (*Ahmadizad & El-Sayed, 2003*; *Ahmadizad &*

*El-Sayed, 2005*). Changes in plasma volume during exercise is caused, among others, by the increase in hydrostatic pressure in capillaries, and an increase in osmotic pressure in the muscle tissue. It is a result of accumulation of osmotically active metabolites in working muscles, especially lactate (*Connes et al., 2013*). The post-exercise lactate concentration in our athletes was above 10 mmol $\cdot$ l$^{-1}$.

## Hematological parameters during restitution

It is worth emphasizing that despite the changes occurring during exercise, we observed a relatively fast return of RBC parameters and Hct to baseline values during 30-min recovery. The same pattern of changes in blood rheological parameters was earlier reported after a single session of heavy resistance training (*Ahmadizad & El-Sayed, 2005*). It was noted that the rise in RBC count, Hb, and Hct was transient and their values also returned to resting levels after 30 min of recovery. The results are consistent with previous findings showing that short maximal running does not alter the increased erythropoiesis in trained athletes (*Nader et al., 2018*). In our study, as it was in other studies (*Ahmadizad & El-Sayed, 2005*; *Nader et al., 2018*), we noted no change in MCV during exercise or the restitution phase. This indicates the lack of erythrocyte shrinkage (*Nader et al., 2018*). Strenuous running exercise may affect hematological variables due to RBC damage through repeated foot strike impact (*Eichner, 1985*; *Telford et al., 2003*), but it seems that highly trained athletes have adaptation mechanisms that protect RBC (*Mairbäurl, 2013*; *Pospieszna et al., 2019*; *Powers, Nelson & Hudson, 2011*; *Sentürk et al., 2005*). It has been proven that young erythrocytes are more flexible and have higher deformability than old ones (*Tomschi et al., 2018*). This positively affects the improvement of microcirculation. Competitive athletes possess a greater proportion of younger RBC, characterized by more efficient metabolism and higher energy status than recreationally trained ones (*Mairbäurl, 2013*; *Pospieszna et al., 2023*; *Pospieszna et al., 2021*; *Pospieszna et al., 2019*). It was also shown that regular intense training increases the activity of antioxidant enzymes and improves the antioxidant defense system of erythrocytes (*Powers, Nelson & Hudson, 2011*; *Sentürk et al., 2005*).

## Hematological parameters and intensity of exercise

The intensity training zones are usually defined by physiological determination of the first and second ventilatory 'turnpoints'. Training loads can be divided into: low-intensity loads (below VT1), moderate-intensity loads (between VT1 and second VT2), and high-intensity loads (above the VT2) (*Seiler & Tønnessen, 2009*). In this study, no significant changes were recorded in hematological parameters during the low-intensity phase. Throughout the incremental exercise test, RBC, Hb, and Hct increased significantly between VT1 and VT2. *Robert et al. (2020)* showed that 40 km of low-speed running (6.1 $\pm$ 1.6 km $\cdot$ h$^{-1}$) did not cause changes in post-exercise Hct, MCV, and MCHC in endurance trained athletes. Furthermore, low plasma free Hb levels after low-intensity running indicated a low percentage of hemolysis. Our findings are contrasting to the results obtained in sedentary young adults indicating that RBC, Hb, and Hct increased after moderate intensity exercise, but not after high-intensity intermittent exercise (*Findikoglu et al., 2014*). These inconsistencies may be due to differences in a number of adaptations in the body, including the circulatory system with both major and microcirculatory vessels (*Green et al., 2017*).

## Hematological parameters in athletes from various sports specializations

The research findings on the association between the type of training loads and the level of hematological parameters are inconsistent. In previous studies, endurance athletes had significantly higher Hb (*Pospieszna et al., 2019*; *Zieliński & Kusy, 2012*) and Hct (*Pospieszna et al., 2019*) levels throughout the training season compared to sprinters, but strength athletes had higher Hb, Hct, and RBC than endurance athletes (*Schumacher et al., 2002*). Our results are consistent with studies that showed no difference in hematological parameters between elite endurance-trained and other athletes (*Ciekot-Sołtysiak et al., 2018*; *Malczewska-Lenczowska et al., 2013*; *Zarebska et al., 2018*). Also, in our earlier study (*Ciekot-Sołtysiak et al., 2018*), sprinters and endurance athletes did not significantly differ in most hematological parameters during the annual training cycle. Based on the obtained results, we noticed that the pattern of changes was the same in all athletes and depended on the intensity of exercise. At lower exercise intensity we did not observe changes in RBC parameters and Hct. Only high-intensity exercise an increase in RBC, Hct, and Hb. The most RBC parameters increased gradually during exercise, reaching the highest value during maximum exertion, and then decreased to baseline. The pattern of changes in Hct and RBC parameters during exercise of increasing intensity seems to be universal, regardless of the athlete's physiological profile.

## Strengths and limitations

Previous studies describing changes in hematological parameters during exercise were based only on blood samples taken before and after exercise, ignoring changes related to the intensity of exercise. The strength of this article is that by combining assessments before, during and after exercise, we obtained a comprehensive picture of exercise induced change in RBC parameters. The limitation of our research is that we have not evaluated the athletes' hydration level and sweating rate during exercise. Another limitation is that we focused only on male athletes. It is difficult to assess whether the conclusions from our research could also apply to women. There are no studies on changes in hematological parameters during exercise in female athletes. Premenopausal women have lower values of hematological parameters than men matched for age and race (*Murphy, 2014*), but data regarding sex differences in hematological parameters in highly trained athletes is not consistent. The majority of studies showed lower resting values of hematological parameters in female compared to male athletes (*Malczewska-Lenczowska et al., 2013*; *Sharpe et al., 2002*; *Trinschek et al., 2023*). *Tomschi, Bloch & Grau (2018)* found no gender differences in Hct and RBC parameters if sports specialization was taken into account. Other research results also suggest that the type of training may influence the resting level of blood parameters in women. Endurance-trained women had higher Hct and Hb than women practicing esthetic sports, power sport, team sports (*Sharpe et al., 2002*), and combat sports (*Malczewska-Lenczowska et al., 2013*). On the other hand, there were no significant difference in hematological parameters between female athletes representing sprint and endurance disciplines (*Trinschek et al., 2023*), and combat, team sport, racket, and strength sports (*Tomschi, Bloch & Grau, 2018*). Inconclusive results

regarding hematological parameters in female athletes suggest the need for further research in this area. Among the limitations, we should also highlight that the conclusions of our research apply only to highly trained athletes.

## CONCLUSIONS

RBC, Hb, and Hct increase significantly during incremental physical exercise, and then return to resting values within 30 min upon test termination. The general pattern of exercise-induced changes in RBC parameters and Hct is universal regardless of the athlete's physiological profile. On the other hand, the changes in Hct and RBC parameters are proportional to the intensity of exercise during the progressive test. The increase in hemoglobin concentration associated with the intensity of exercise is most likely an adaptation to the greater demand of tissues, mainly skeletal muscles, for oxygen.

## ACKNOWLEDGEMENTS

We thank the coaches of the Polish national teams as well as athletes for full cooperation.

### Funding
This work was supported by funding from the National Science Centre Poland under grant OPUS 14 number 2017/27/B/NZ7/02828. The funders had no role in study design, data collection and analysis, decision to publish, or preparation of the manuscript.

### Grant Disclosures
The following grant information was disclosed by the authors:
The National Science Centre Poland: OPUS 14 number 2017/27/B/NZ7/02828.

### Competing Interests
The authors declare there are no competing interests.

### Author Contributions
- Monika Ciekot-Sołtysiak conceived and designed the experiments, performed the experiments, analyzed the data, prepared figures and/or tables, authored or reviewed drafts of the article, and approved the final draft.
- Krzysztof Kusy conceived and designed the experiments, performed the experiments, analyzed the data, authored or reviewed drafts of the article, and approved the final draft.
- Tomasz Podgórski performed the experiments, analyzed the data, authored or reviewed drafts of the article, and approved the final draft.
- Barbara Pospieszna performed the experiments, analyzed the data, authored or reviewed drafts of the article, and approved the final draft.
- Jacek Zieliński conceived and designed the experiments, performed the experiments, analyzed the data, authored or reviewed drafts of the article, and approved the final draft.

## Human Ethics

The following information was supplied relating to ethical approvals (i.e., approving body and any reference numbers):

The project was approved by the Ethics Committee at the Karol Marcinkowski Poznan University of Medical Sciences (Decision No 1252/18).

## Data Availability

The raw data are available in the Supplemental File and at RepOD: Ciekot-Sołtysiak, Monika; Kusy Krzysztof; Podgórski Tomasz; Pospieszna Barbara; Zieliński Jacek, 2023, "Raw data connected with article entitled: Changes in red blood cell parameters during incremental exercise in highly trained athletes of different sport specializations". Available at https://doi.org/10.18150/BB34UL, RepOD, V1.

## Supplemental Information

Supplemental information for this article can be found online at http://dx.doi.org/10.7717/peerj.17040#supplemental-information.

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
