# Peer review of "Changes in red blood cell parameters during incremental exercise in highly trained athletes of different sport specializations"

_PeerJ, doi:10.7717/peerj.17040_

## Round 0.1 · original submission · Major Revisions

Dear Authors:

Thank you for thinking of PeerJ to submit your manuscript. Some improvements must be made to be considered. Please attend to the reviewers´ comments.

Regards.

Dr. Manuel Jiménez

·

Basic reporting

BASIC REPORTING:
• The study topic is interesting and innovative, however, if the writers make use of sub headings to simplify it and more reader – friendly. Moreover, blood samples are taken before and after exercise only. No mention of changes during the course of exercise? My suggestion is that by combining assessments before, during and after exercise, a comprehensive picture will emerge which will aid in tailoring training programs, identifying areas for improvement and gauging overall health and performance adaptations.
• Line 29-30: mentions that there were no significant exercise-induced differences in red blood cells parameters between athletic groups. Later on the statement is different in conclusion. Please clarify this confusion.
Moreover, what is the difference between red blood cells parameters and hematological parameters?
• Sports terminologies need to be clarified, for example, endurance and mixed-trained athletes (Line 71 – 72), competitive athletes of different specialties (Line 120), Triathletes (Line 193) etc
• Why female gender is being excluded? (A limitation mentioned by authors – Line 359)
• Why goal keepers were excluded?
• Line 135: Improve the sentence (use of two negative?)

Experimental design

• No issues found, ANOVA statistical method to analyze the differences among different groups is justified in my opinion.

Validity of the findings

• No issues found at a glance. However, cross-validation using different data sources may be used to see if results remain consistent. This will ensure that findings are not dependant on a specific data set approach
• Seek input from Sports personalities, e.g. Physiotherapists, to get diverse perspectives and insights
• Conduct longitudinal studies and followup, if possible.

Additional comments

• Clinical implications for this study is not mentioned esp. in the Conclusion. For example, Hb levels rise to enhance oxygen carrying capacity of blood to meet the increased demand for oxygen transport during exercise contributing to improved aerobic performance

Reviewer 2 ·

Basic reporting

General considerations:
It is important to carry out a general review of the text regarding wording and verbal adequacy.
The text is relatively clear and sufficient for the reader to understand.
As for the individuals studied, they have different physical profiles and different sports, which can influence the results and cause bias in the analyses. With this division, the number of athletes in each group is small and can influence the statistical analysis.
As for plasma volume, hydration control is important and was not considered, as well as sweating during physical activity, which is different individually due to hormonal issues and exercise intensity.
I believe that the time between analyzes was small for the physiological response.

Experimental design

For this theme is important responde:
As for the individuals studied, they have different physical profiles and different sports, which can influence the results and cause bias in the analyses. With this division, the number of athletes in each group is small and can influence the statistical analysis.
As for plasma volume, hydration control is important and was not considered, as well as sweating during physical activity, which is different individually due to hormonal issues and exercise intensity.
I believe that the time between analyzes was small for the physiological response.

Validity of the findings

It is important to review the topics highlighted above and explain the interferences in the analyses.

Additional comments

The article is interesting and important for exercise physiology. However, it is necessary to clarify the highlighted points to avoid interpretative bias.

---

## Round 0.2 · accepted · Accept

Dear Author: it is a pleasure to inform you that your manuscript has been accepted for publication.

Congratulations

Dr. Manuel Jiménez